# Kangaroo mother care knowledge, attitude, and practice among nursing staff in a hospital in Jakarta, Indonesia

**Asri Adisasmita** [1]*, **Yulia Izati**[2], **Septyana Choirunisa**[2], **Hadi Pratomo**[3], **Luzy Adriyanti**[4]

**1** Department of Epidemiology, and Kangaroo Mother Care Research Project, Faculty of Public Health, Universitas Indonesia, Depok, Indonesia, **2** Kangaroo Mother Care Research Project, Faculty of Public Health, Universitas Indonesia, Depok, Indonesia, **3** Department of Health Education and Behavioural Sciences, and Kangaroo Mother Care Research Project, Faculty of Public Health, Universitas Indonesia, Depok, Indonesia, **4** Koja District General Hospital, North Jakarta, Jakarta Province, Indonesia

* aadsmt237@gmail.com, adisasa@ui.ac.id

**Data Availability Statement:** The datasets generated and/or analyzed in this study are not publicly available because they are property of Universitas Indonesia and Koja District General

## Abstract

### Background

Kangaroo mother care (KMC) has been proven to decrease rates of morbidity and mortality among premature and low-birth-weight infants. Thus, this study aimed to obtain baseline data regarding KMC knowledge, attitudes, and practices (KAP) among nursing staff caring for mothers and newborns in a hospital in Indonesia.

### Methods

This cross-sectional study included 65 participants from three hospital wards at Koja District Hospital, North Jakarta. Participants included 29 perinatal ward nurses, 21 postnatal ward nurses and midwives, and 15 labor ward midwives. Data on KAP of KMC were collected using a self-administered questionnaire with closed-ended questions. Each questionnaire can be completed in approximately 1 hour.

### Results

Among the included nursing staff, 12.3% (8/65) were determined to have received specific training on KMC, whereas 21.5% (14/65) had received more general training that included KMC content. About 46.2% of the nursing staff had good knowledge concerning KMC, 98.5% had good knowledge of KMC benefits, and 100% had a positive attitude toward KMC. All perinatal ward nurses had some experience assisting and implementing KMC. Some KAP that were observed among the nursing staff included lack of knowledge about the eligible infant weight for KMC and weight gain of infants receiving KMC, lack of education/training about KMC, and concerns regarding necessary equipment in KMC wards.

### Conclusions

This study identified several issues that need to be addressed, including knowledge of feeding and weight gain, workload, incubator use, and the need for well-equipped KMC wards.

Hospital; however, they may be available from the corresponding author on reasonable request. The authors confirm that there is no special access privileges to the data others would not have. Data can be requested to: 1. Research and Community Engagement Unit, Faculty of Public Health, Universitas Indonesia (Dr. Doni H. Ramadan, email: fkmui@ui.ac.id) 2. Research and Development Unit, Koja District General Hospital (Ms. Titin Windarti, email: info@rsudkoja.jakarta.go.id)

**Funding:** This study was made possible by the generous support of the American people through the PEER (Partnerships for Enhanced Engagement in Research) Program. The program is supported by USAID and implemented by the U.S. National Academies of Sciences. HP (as the principal investigator of this study) received funding under Sponsor Grant Award Number: AID-OAA-A-11-00012. NAS URL: http://www.nasonline.org/ USAID PEER Program https://www.usaid.gov/what-we-do/GlobalDevLab/international-research-science-programs/peer The contents of this study are the sole responsibility of the authors and do not necessarily reflect the views of the USAID or the United States Government. The publication of this study was made possible by support of Universitas Indonesia (UI research grant 2019: PENG-1/UN2.R3.1/PPM.00/2019) including funding for publication. The funders had no role in the study design; data collection and analysis, decision to publish, or preparation of the manuscript.

**Competing interests:** The authors have declared that no competing interests exist.

We recommend that hospitals improve their nursing staff's knowledge of KMC and establish well-equipped KMC wards.

## Introduction

Worldwide, nearly 14.8 million babies are born prematurely. In 2014, Indonesia ranked fifth in the world for the number of preterm births (527,672 infants), which comprised 3.5% of preterm births globally for that period [1]. This is important because preterm infants have considerably higher mortality rates than full-term babies [2]. Studies have shown that kangaroo mother care (KMC) is safe and effective for managing low-birth-weight (LBW) and preterm babies, and it contributes to decreased mortality rates of preterm infants in both low- and high-income countries [2]. Lawn et al. reported that KMC was associated with an approximately 50% decrease in deaths among LBW infants weighing less than 2,000 grams at birth [3]. Moreover, unlike the conventional method of care (incubator), it has been reported that KMC reduces the incidence of severe infection (such as sepsis), nosocomial infection, hypothermia, severe morbidity, lower respiratory tract infection, and prolonged hospital stay. Compared to infants receiving conventional care, those treated with KMC showed more stable body temperature; increased body weight, length, and head circumference; improved breastfeeding; and stronger mother–child bonding [3, 4].

Although KMC was introduced in Indonesia in the 1990s, it was not until much later that the Ministry of Health of the Government of Indonesia (GoI) has started to intensively promote KMC in several hospitals throughout the country. With the assistance of the United States Agency for International Development (USAID) and through the Health Service Program (2006–2012), the GoI initiated the KMC program by sending healthcare personnel to attend KMC training in South Africa. Subsequently, the KMC method was implemented across several private and public hospitals [5]. However, KMC implementation did not progress as expected due to lacking guidelines for KMC implementation and standard operating procedures (SOPS), absence of routine supervision and mentoring, and need for awareness/knowledge in healthcare workers (nursing staff) regarding the importance of KMC for the care of LBW/preterm babies. Those factors may have affected nursing staff knowledge, attitude, and practices (KAP) concerning KMC. Thus, good KAP about KMC are necessary to ensure good implementation of KMC in hospitals. Studies have shown that lack of training, which affected KAP of KMC, substantially hindered KMC adoption among nurses, especially in low- and middle-income countries [6]. Zhang et al. (2018) identified inadequate formal education among nurses as a substantial barrier to the implementation of KMC in China [7]. Even informal education about KMC provided improved knowledge of KMC and its benefit among nurses who could support KMC care [2, 7, 8].

To date, information on KAP of KMC among nursing staff in Indonesia remains limited. Therefore, this study aimed to describe the KAP of nursing staff concerning KMC in a hospital setting, serving as a baseline for larger intervention studies as part of efforts to improve the KAP of KMC among nursing staff.

## Materials and methods

This descriptive study examining KAP of KMC among nursing staff was conducted as part of a baseline research project on KMC implementation at Koja District Hospital in North Jakarta, Indonesia, and it was funded by the USAID Partnerships for Enhanced Engagement in

Research (PEER) program. The study is part of a larger research project that used quantitative and qualitative methods, but this part only focused on the quantitative part of the research. The complete baseline research project included assessment of neonatal morbidity, mortality, and KMC implementation; KAP of KMC among hospital nursing staff; and KAP of KMC among primary health center staff. It also included formative research that explored factors needed to strengthen and enhance KMC implementation in the hospital. Factors included refresher training, facilitative supervision, hospital policies, SOPs for KMC implementation, and the needed supporting facilities to provide input for the development of an intervention package to improve KMC implementation in the hospital. This current study included an analysis of baseline conditions regarding the KAP of KMC among all nurses supporting KMC care before the implementation of the intervention package. This baseline study aimed to provide input for designing a training as part of the intervention to improve KMC care.

The study site, Koja District Hospital, was a secondary-level hospital with a 16-bed neonatal intensive care unit (NICU) and a 40-bed perinatal ward that received perinatal referrals from hospitals near Jakarta. There were no rooms or beds specifically designated for KMC and no written formal KMC SOPs signed by the hospital director. Between June and December 2015, approximately 3,040 neonates were born at or admitted to the hospital. Among those, 412 had LBW (13.5%), with 85% born at the hospital and the remaining admitted after birth. The death rate among LBW infants was recorded to be at 0.9% of the total number of neonatal deaths per total number of those born at or admitted to the hospital during the same period [9].

All nursing staff at Koja District Hospital who provides in-hospital neonatal care (65 respondents) were included: 29 perinatal ward and NICU nurses, 21 postnatal ward nurses and midwives, and 15 labor ward midwives. The perinatal ward is where neonates are hospitalized, the labor ward is where women deliver their babies, and the postnatal ward is where mothers are transferred after giving birth to a healthy baby. Nursing staff working at an antenatal care (ANC) clinic and those from other hospitals who referred neonates to Koja District Hospital were excluded from the baseline assessment. The hospital nursing staff of each ward was divided into three work shifts. For example, nine nurses provide care for 40 and 16 neonates in the perinatal ward and NICU during each shift, respectively, yielding a ratio of 9 nurses to 56 infants, assuming the usual full occupancy.

Included nursing staff signed a written informed consent form prior to study enrollment. None of them refused to participate in the study. Thereafter, they completed a self-administered questionnaire that included questions related to general knowledge, benefits, attitude, and practices associated with KMC. Data were collected in March 2016. Before data collection, a pre-test of the questionnaire was conducted with nursing staff from another hospital similar to the study site.

The questionnaire consisted of 24 questions on general knowledge, 20 questions on benefits, 12 questions on attitude, and 6 questions on practices associated with KMC. To measure general knowledge, benefits, and practices related to KMC, closed-ended questions (adapted from El-Nagar et al. and KMC Facilitator's Guide ACCESS) were used [10, 11]. For every question on general knowledge and benefits of KMC, a correct response was scored as 1 and incorrect as 0. To measure attitude toward KMC, questions from validated tools were adapted, and a 5-point Likert scale (ranging from "strongly disagree" to "strongly agree") was used. During analysis, the Likert scale was converted to scores, with 60 being the highest and 12 being the lowest (based on 12 questions and a 5-point Likert scale). A maximum score of 75% or more (i.e., 45 or more) reflected a good attitude toward KMC. Similarly, the same approach was applied to categorize general knowledge and benefits of KMC. Maximum scores for general knowledge and benefits of KMC were 24 and 20, respectively. General knowledge, knowledge

**Table 1. Nursing staff characteristics.**

| Characteristics | Total | Perinatal ward (n = 29) | Labor ward (n = 15) | Postnatal ward (n = 21) | p-value |
|---|---|---|---|---|---|
| | n | n (%) | n (%) | n (%) | |
| **Age (years)** | | | | | |
| <30 | 38 | 21 (72.4) | 5 (33.3) | 12 (57.1) | 0.044* |
| >30 | 27 | 8 (27.6) | 10 (66.7) | 9 (42.9) | |
| **Years of experience working in the current unit (years)** | | | | | |
| <10 | 58 | 27 (93.2) | 11 (73.3) | 20 (95.2)[a] | 0.023* |
| >10 | 6 | 2 (6.8) | 4 (26.7) | 0 (0.0) | |
| **Highest education attained** | | | | | |
| Diploma | 55 | 26 (89.7) | 13 (86.7) | 16 (76.2) | 0.415 |
| Bachelor | 10 | 3 (10.3) | 2 (13.3) | 5 (23.8) | |

[a]Percentages that do not add up to 100% was due to missing data (i.e., one respondent from the postnatal ward did not provide an answer for years of experience working in the current unit).

*p-value < 0.05.

on benefits, and attitude score were grouped into four categories (scores 0 to 25 = very low, 25 to below 50 = low, 50 to below 75 = moderate, and 75 to 100 = good/high).

Data collected were entered and analyzed using statistical software. We used descriptive statistics (frequencies, proportions, mean, and median) to summarize knowledge scores and chi-squared or Fisher exact tests if the expected value was less than 5 to measure the associations between two variables. P-values < 0.05 were considered statistically significant. This study was approved by the Ethics Committee of Faculty of Public Health, Universitas Indonesia (Registry number 230/H2.F10/PPM.00.02/2015).

## Results

### Nursing staff characteristics

Characteristics of the nursing staff in this study are presented in Table 1. Nursing staff in the perinatal–NICU and postnatal wards were determined to be younger (<30 years of age) than those in the labor ward. Most of the nursing staff (93.2%, 73.3%, and 95.2% in the perinatal, labor, and postnatal wards, respectively) had been working for < 10 years in their current unit. At least 75% of the staff in all three wards had nurse or midwife diplomas (3 years of education).

### Kangaroo mother care (KMC) training

Most study participants had never attended KMC-specific training (87.7%) or other training that included KMC content (78.5%). Interestingly, a higher proportion of nursing staff in the labor ward than in the perinatal ward attended training that included KMC content (33.3% vs 24.1%, respectively), although the difference was not statistically significant (p = 0.232) (Table 2). Overall, 7.7% of nurses (5/65) had attended both the above-mentioned types of training, 18.5% (12/65) had attended training specific to or including KMC, and 73.8% (48/65) had never attended any KMC-related training.

### General knowledge on KMC among nursing staff

General knowledge on KMC was measured through 24 questions, wherein five of those were answered incorrectly by most of the nursing staff in the labor and postnatal wards. The five

**Table 2. Percentages of nursing staff who had attended kangaroo mother care-related training.**

| Kangaroo mother care (KMC) training | Total | Perinatal ward (n = 29) | Labor ward (n = 15) | Postnatal ward (n = 21) | p-value |
|---|---|---|---|---|---|
| | n | n (%) | n (%) | n (%) | |
| **Attended KMC training** | | | | | |
| [a]Attended both KMC-specific training and training that included KMC | 5 | 2 (6.9%) | 2 (13.3%) | 1 (4.8%) | 0.623 |
| [b]Attended KMC-specific training | 3 | 2 (6.9%) | 0 (0%) | 1 (4.8%) | |
| [c]Attended training that included KMC | 9 | 5 (17.2%) | 3 (20.0%) | 1 (4.8%) | |
| [d]Never attended any training | 48 | 20 (69.0%) | 10 (66.7%) | 18 (85.6%) | |
| **Attended KMC-specific training** | | | | | |
| Yes[(a + b)] | 8 | 4 (13.8) | 2 (13.3) | 2 (9.5) | 0.894 |
| No[(c + d)] | 57 | 25 (86.2) | 13 (86.7) | 19 (90.5) | |
| **Attended training that included KMC** | | | | | |
| Yes[(a + c)] | 14 | 7 (24.1) | 5 (33.3) | 2 (9.5) | 0.208 |
| No[(b + d)] | 51 | 22 (75.9) | 10 (66.7) | 19 (90.5) | |

questions concerned the following: (1) providing nutrition using a small cup, (2) nasogastric tube feeding for LBW infants, (3) early feeding with non-breastmilk, (4) counseling for neonates already at a bodyweight of 2,500 g, and (5) infants gaining adequate weight with the KMC method. Most of the nursing staff (79.3%, 80%, and 100% in the perinatal, labor, and postnatal wards, respectively) incorrectly answered the last question (i.e., weight gain). Nursing staff in the perinatal ward achieved the highest percentage of correct answers (20.7%) for the question on adequate weight gain with the KMC method.

After stratifying participants according to their age, there was no difference observed in the percentage of those with good general knowledge on KMC (63.2% among those aged < 27 years old, compared to 70% among those aged > 27 years old; p = 1.000). The percentage of nursing staff with good general knowledge on KMC was similar between those who had attended training specific to KMC and those who never attended KMC training (50% vs. 68%, respectively; p = 1.000). A higher percentage of perinatal staff who attended training that included KMC content had good general knowledge on KMC compared to those who had never attended such training, although the difference was not statistically significant (71.4% vs. 61.9%, respectively; p = 1.000; Table 3). Scores on general knowledge of KMC were good (score > 75) among 65.5% and 60% nursing staff in the perinatal and labor wards, respectively, but only 9.5% nursing staff in the postnatal ward achieved a good score (Fig 1).

## Nursing staff knowledge on KMC benefits

Most of the nursing staff correctly responded to the 20 questions regarding KMC benefits, which were asked as true or false questions. However, several nursing staff in all three wards incorrectly answered some of these questions. These included questions regarding: (1) using the KMC method for LBW infants kept in an incubator (26.7% and 52.4% of nursing staff in the labor and postnatal ward, respectively, answered incorrectly); (2) disadvantages of KMC implementation with ward hygiene affected because of increased crowdedness (72.4%, 20%, and 23.8% of perinatal–NICU, labor, and postnatal ward nurses, respectively, answered incorrectly), which raised concerns about having a well-equipped KMC ward to support more convenient KMC implementation and prevent cross infection; and (3) whether or not KMC could reduce nursing staff workload (65.5%, 20%, and 4.8% of nursing staff from the perinatal–NICU, labor, and postnatal wards, respectively, stated that KMC increased workload). No difference was observed in the percentage of nursing staff with good knowledge on KMC

**Table 3. Knowledge and attitude toward kangaroo mother care (KMC) according to age and training experience.**

| Parameter | Total | General knowledge on KMC | | | | Knowledge on KMC benefit | | | | Attitude toward KMC | | | |
|---|---|---|---|---|---|---|---|---|---|---|---|---|---|
| | | Low | Moderate | Good/High | p-value | Low | Moderate | Good/High | p-value | Low | Moderate | Good/High | p-value |
| | n | n (%) | n (%) | n (%) | | n (%) | n (%) | n (%) | | n (%) | n (%) | n (%) | |
| **Perinatal ward (n = 29)** | | | | | | | | | | | | | |
| **Age (years)** | | | | | | | | | | | | | |
| <27 | 19 | 0 (0.0) | 7 (36.8) | 12 (63.2) | 1.000 | 0 (0.0) | 0 (0.0) | 19 (100.0) | 0.345 | 0 (0.0) | 0 (0.0) | 19 (100.0) | NA** |
| >27 | 10 | 0 (0.0) | 3 (30.0) | 7 (70.0) | | 0 (0.0) | 1 (10.0) | 9 (90.0) | | 0 (0.0) | 0 (0.0) | 10 (100.0) | |
| **Attended KMC-specific training** | | | | | | | | | | | | | |
| Yes | 4 | 0 (0.0) | 2 (50.0) | 2 (50.0) | 0.592 | 0 (0.0) | 0 (0.0) | 4 (100.0) | 1.000 | 0 (0.0) | 0 (0.0) | 4 (100.0) | NA** |
| No | 25 | 0 (0.0) | 8 (32.0) | 17 (68.0) | | 0 (0.0) | 1 (4.0) | 24 (96.0) | | 0 (0.0) | 0 (0.0) | 25 (100.0) | |
| **Attended training that included KMC** | | | | | | | | | | | | | |
| Yes | 7 | 0 (0.0) | 2 (28.6) | 5 (71.4) | 1.000 | 0 (0.0) | 0 (0.0) | 7 (100.0) | 1.000 | 0 (0.0) | 0 (0.0) | 7 (100.0) | NA** |
| No | 22 | 0 (0.0) | 8 (36.4) | 14 (63.6) | | 0 (0.0) | 1 (4.5) | 21 (95.5) | | 0 (0.0) | 0 (0.0) | 22 (100.0) | |
| **Labor ward (n = 15)** | | | | | | | | | | | | | |
| **Age (years)** | | | | | | | | | | | | | |
| <27 | 2 | 0 (0.0) | 1 (50.0) | 1 (50.0) | 1.000 | 0 (0.0) | 0 (0.0) | 2 (100.0) | NA** | 0 (0.0) | 0 (0.0) | 2 (100.0) | NA** |
| >27 | 13 | 0 (0.0) | 5 (38.5) | 8 (61.5) | | 0 (0.0) | 0 (0.0) | 13 (100.0) | | 0 (0.0) | 0 (0.0) | 13 (100.0) | |
| **Attended KMC-specific training** | | | | | | | | | | | | | |
| Yes | 2 | 0 (0.0) | 2 (100.0) | 0 (0.0) | 0.143 | 0 (0.0) | 0 (0.0) | 2 (100.0) | NA** | 0 (0.0) | 0 (0.0) | 2 (100.0) | NA** |
| No | 13 | 0 (0.0) | 4 (30.8) | 9 (69.2) | | 0 (0.0) | 0 (0.0) | 13 (100.0) | | 0 (0.0) | 0 (0.0) | 13 (100.0) | |
| **Attended training that included KMC** | | | | | | | | | | | | | |
| Yes | 5 | 0 (0.0) | 5 (100.0) | 0 (0.0) | 0.002* | 0 (0.0) | 0 (0.0) | 5 (100.0) | NA** | 0 (0.0) | 0 (0.0) | 5 (100.0) | NA** |
| No | 10 | 0 (0.0) | 1 (10.0) | 9 (90.0) | | 0 (0.0) | 0 (0.0) | 10 (100.0) | | 0 (0.0) | 0 (0.0) | 10 (100.0) | |
| **Postnatal ward (n = 21)** | | | | | | | | | | | | | |
| **Age (years)** | | | | | | | | | | | | | |
| <27 | 12 | 1 (8.3) | 9 (75.0) | 2 (16.7) | 0.435 | 0 (0.0) | 0 (0.0) | 12 (100.0) | NA** | 0 (0.0) | 0 (0.0) | 12 (100.0) | NA** |
| >27 | 9 | 1 (11.1) | 8 (88.9) | 0 (0.0) | | 0 (0.0) | 0 (0.0) | 9 (100.0) | | 0 (0.0) | 0 (0.0) | 9 (100.0) | |
| **Attended KMC-specific training** | | | | | | | | | | | | | |
| Yes | 2 | 0 (0.0) | 2 (100.0) | 0 (0.0) | 0.771 | 0 (0.0) | 0 (0.0) | 2 (100.0) | NA** | 0 (0.0) | 0 (0.0) | 2 (100.0) | NA** |
| No | 19 | 2 (10.5) | 15 (79.0) | 2 (10.5) | | 0 (0.0) | 0 (0.0) | 19 (100.0) | | 0 (0.0) | 0 (0.0) | 19 (100.0) | |
| **Attended training that included KMC** | | | | | | | | | | | | | |

*(Continued)*

**Table 3.** (Continued)

| Parameter | Total | General knowledge on KMC | | | | Knowledge on KMC benefit | | | | Attitude toward KMC | | | |
|---|---|---|---|---|---|---|---|---|---|---|---|---|---|
| | | Low | Moderate | Good/High | p-value | Low | Moderate | Good/High | p-value | Low | Moderate | Good/High | p-value |
| | n | n (%) | n (%) | n (%) | | n (%) | n (%) | n (%) | | n (%) | n (%) | n (%) | |
| Yes | 2 | 0 (0.0) | 2 (100.0) | 0 (0.0) | 0.771 | 0 (0.0) | 0 (0.0) | 2 (100.0) | NA** | 0 (0.0) | 0 (0.0) | 2 (100.0) | NA** |
| No | 19 | 2 (10.5) | 15 (79.0) | 2 (10.5) | | 0 (0.0) | 0 (0.0) | 19 (100.0) | | 0 (0.0) | 0 (0.0) | 19 (100.0) | |

*p-value <0.05.

**p-values cannot be calculated for all participants included in one category.

according to age group. All nursing staff with any training involving KMC content showed good knowledge of KMC benefits, while those with no KMC training had slightly lower knowledge (95.2%) (Table 3). Most nurses in all wards had good/high scores (scores > 75) for knowledge (Fig 1).

## Attitude toward KMC

Attitude of the nursing staff toward KMC was assessed through 12 questions. When asked about KMC implementation among infants weighing 1,000–1,800 g, 40% of respondents from the labor ward were not in favor of KMC implementation. No differences were observed after stratifying respondents' scores according to age group and training experience, and 100% of nurses had a good attitude (score > 75) about KMC (Table 3). These results remained consistent in all wards (Fig 1).

## Practices

KMC practices by nursing staff were measured based on two parameters: (1) educating mothers/fathers/families on KMC implementation for LBW infants and (2) assisting with KMC

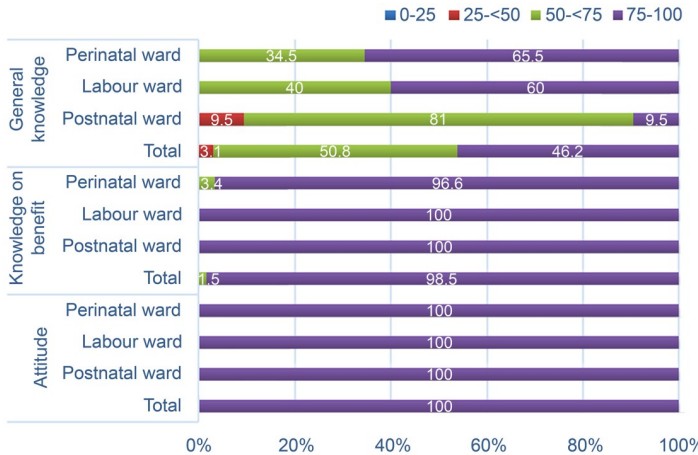

**Fig 1. Percentages of correct answers on knowledge and attitude toward kangaroo mother care.** General knowledge, knowledge on benefits, and attitude scores were grouped into four categories (scores 0 to 25, 25 to below 50, 50 to below 75, and 75 to 100). The figure represents the percentages of nurses based on those four score categories, with a score of 75–100 representing good knowledge or attitude.

implementation. All nursing staff in the perinatal ward stated that they educated and assisted mothers/fathers/families to implement KMC for their infants. For nursing staff working in labor and postnatal wards, 9 of 36 also had the opportunity to educate and assist mothers in KMC.

## Discussion

Several studies have shown that formal and informal KMC education among nursing staff can substantially increase the success of KMC implementation. Unsuccessful KMC implementation is due to, among other factors, uncertainty about including or excluding neonates for KMC, which decreases confidence in implementing KMC. Despite being a single-center study, our results could be generalizable to the status of KAP of KMC among all hospital nursing staff in Indonesia, where such data remain scarce. Our findings showed that only 12.3% of all nurses had attended KMC-specific training and 21.5% attended other training that included KMC content. Zhang et al. (2018) found that KMC-specific training increased the confidence of neonatal nurses in KMC, thus promoting its implementation [7]. Bergh et al. mentioned that despite enthusiasm about participating in KMC training, the distance to the training site and staff shortages within a hospital could be the reason why a limited number of staff attended KMC training [12]. Considering that only a few nurses in Koja District Hospital had received KMC-specific training, we recommend that all nurses who provide mother and infant care at Koja District Hospital receive KMC-specific training to increase knowledge and confidence in implementing KMC. This is essential for vulnerable infants, such as those born preterm or LBW. This baseline study was conducted to obtain information to be incorporated into in-house training intervention at Koja District Hospital. Another solution to be considered is to consistently include KMC content in the nurse/midwife training curriculum [13].

Our findings show that most perinatal–NICU ward nurses had already been implementing KMC and educating parents on KMC, despite their lack of KMC training. Providing actual training to the nursing staff shows promise of substantially improving KMC implementation at Koja District Hospital. Also, the lack of training could result in conflicting knowledge on the timing and duration of KMC [14, 15], which could lead to adverse consequences such as mortality [16] especially among less stable infants.

General knowledge on KMC was surprisingly low among most midwives and nurses working in the postnatal ward. Postnatal ward nursing staff must have adequate knowledge on KMC to ensure its implementation among stable infants born in the hospital, including those rooming-in with the mother and those transferred from the perinatal–NICU ward whose mothers remain hospitalized. The current policy in the study hospital, however, does not support KMC implementation in the postnatal ward. Nonetheless, studies have shown that implementing KMC in the postnatal ward could reduce the length of hospital stay among more stable LBW infants, thereby reducing the cost of care [17–19].

Those who had attended KMC trainings had consistently better general knowledge of KMC and its benefits. However, there was no statistically significant difference in the attitude toward KMC between those who had and had not received KMC training. Our questionnaire was based on previous studies that collected data through interviews [10, 11]. However, the self-administered method of collecting data in our study could have hindered our ability to detect differences in attitude toward KMC, as nurses could discuss how to respond to questions while completing the questionnaire.

This present study also showed that certain KAP of KMC, including knowledge and concepts of feeding and weight gain, workload, and incubator use, are in urgent need of improvement to achieve better KMC implementation. One issue that needs to be highlighted in

training is knowledge regarding feeding and weight gain, which is central to the success of KMC. An interventional study in India among medical students revealed that education was the most effective method to improve knowledge on infant feeding [20].

Another noteworthy finding of this study was that KMC did not reduce workload. These results were consistent with those presented in other studies, which found that KMC increased nurses' workload [6, 21, 22]. A nurse may not have time for training to improve their knowledge and understanding of KMC because of their already high workload, leaving them with partial comprehension regarding KMC. A pre- and post-intervention study in a neonatal unit in Sweden (2008 and 2010) found that before the intervention, several nurses indicated that KMC increased their workload, but others stated that it did not. However, no staff members expressed concern about increased workload with KMC after the intervention. In fact, several staff members stated that KMC decreased nurses' workload because parents learned to care for their infants [23]. The Swedish study also concluded that knowledge regarding KMC was especially important, in addition to ensuring the availability of adequate facilities [23]. That study indicated that an effective intervention could lead to a better atmosphere for implementing KMC. Hence, concerns regarding increased workload with KMC in the current study could be addressed through effective intervention. Furthermore, a parallel study of our KMC PEER project using a qualitative approach revealed that some nurses believed that KMC would decrease workload because the mother of a LBW infant is key in KMC implementation. However, to educate and support the mother/family in KMC, the nursing staff needs formal training to improve their skill and confidence in supporting KMC [24]. Adequate education on skin-to-skin contact, feeding position, and infant positioning by nursing staff will lead to excellent quality of KMC among mothers/parents/family. Excellent quality of KMC could promote faster weight gain and thermostability, leading to shorter hospitalization and lower morbidity rates among LBW infants [19]. Proper implementation of KMC could lead to reduced nursing staff workload.

Apart from the lack of understanding of KMC, hospital management often does not prioritize KMC. Provision of facilities and equipment needed to implement KMC, such as KMC beds or couches or a designated ward, could enable the successful and continuous implementation of KMC. The larger study found that the lack of equipment, facilities, and supporting policy were among the reasons why nurses and midwives from wards other than the perinatal ward never implemented KMC. In addition, pediatricians, nursing staff, and management expressed the need of having interventions, such as training, supportive facilitation, and supporting policy, besides facilities and equipment availability to improve KMC implementation [9].

Responses to our questionnaire indicated that most of the nursing staff were hesitant to implement KMC while infants were in an incubator, which was consistent with the qualitative results [24], despite studies showing that KMC could improve cardiorespiratory function, promote temperature stability, and prevent infection in addition to its beneficial effect on sleep patterns and breastfeeding [2]. Therefore, adequate education to address this hesitancy could improve the implementation of KMC with infants in incubators. However, in addition to adequate education, adequate equipment/facilities as well as SOPs and hospital policy supporting KMC implementation are essential for successful KMC in the hospital.

This present study clearly showed that training opportunities should be created to tackle misconceptions and gaps in knowledge of KMC. All nursing staff involved in maternal and infant care (i.e., those in ANC clinics; labor, perinatal, and postnatal wards) should receive adequate training on KMC, even if not directly involved in its implementation. This way, all nursing staff in related wards will be equipped with KMC knowledge that they could use to assist and advise expectant mothers about KMC. An interventional study on KMC knowledge

among pregnant women in an ANC clinic in India showed a significant improvement in knowledge regarding the period when KMC should be started, duration of each session, frequency of KMC, and clothing and positioning of the mother and baby [25]. This illustrates the importance of providing KMC education to pregnant women during ANC visits, given that expectant mothers may be receptive to the concept of KMC. For effective ANC education, nursing staff must have adequate KAP of KMC through effective training.

A study in Norway found that implementing skin-to-skin care among preterm infants in the labor ward may be feasible and safe [26], indicating that the labor ward staff should also be equipped with KMC knowledge. However, implementing KMC in the labor ward at Koja District Hospital would require a preliminary study to assess its feasibility and safety, especially among preterm infants less than 34 weeks of gestation. Nevertheless, midwives on the labor ward can introduce KMC education for expectant mothers of preterm and LBW infants despite not being directly involved in KMC implementation.

Studies have shown that KMC promotes emotional stability, successful breastfeeding, and reduced neonatal morbidity [27]. Nurses/midwives should persuade and educate mothers in postnatal wards about KMC, especially those having LBW infants in stable condition. Bergh et al. [28] suggested that KMC education and training should include all obstetric and neonatal staff members. Indeed, our findings suggest that educational intervention should involve the staff of not only the perinatal ward but also the labor and postnatal wards.

Although all perinatal ward nurses included in this study stated that they had experience in implementing and supporting KMC, the quality of its implementation remained undetermined, with potential for improvement following appropriate intervention. Education has been viewed as an essential tool in improving nurses' knowledge and skills in facilitating KMC [14, 29, 30]. Nevertheless, even nursing staff who had received training may still need time to become comfortable with the method [30, 31]. On-site training could provide additional intervention that would result in the most successful implementation of KMC [32, 33]. Moreover, collaboration among healthcare workers with shared goals and team commitments in which inexperienced nurses are partnered with those experienced in KMC can also be helpful [31, 34, 35]. Overall, KMC should be implemented among eligible LBW infants given that it improves weight gain and growth, protects from sepsis and hypothermia, and promotes breastfeeding, thereby playing an integral role in decreasing mortality and morbidity. To effectively implement KMC, the associated KAP among all related hospital nursing staff (including those in the ANC unit) should be improved. This can be attained through interventions involving education, on-site training, and mentoring. The successful implementation of KMC requires relevant education of nurses, education and support of mothers by nursing staff, monitoring of KMC implementation by nurses, identification of institution-specific barriers, and implementation of institution-specific strategies to overcome these barriers.

Our study showed that most nursing staff in the labor and postnatal wards had not received any KMC training. It is worth noting that midwives in the labor ward are in a good position to persuade mothers of LBW infants to implement KMC as a continuation of early breastfeeding initiation (provision of mother's breast milk to infants within 1 hour of birth) [36]. Nursing staff in the postnatal ward can also persuade mothers of premature infants to implement KMC through educating them on the benefits of KMC (e.g., preventing hypothermia). Thus, including nurses/midwives in wards other than the perinatal ward in the training intervention is justified considering their opportunity to educate mothers with LBW infants regarding KMC.

One strength of this present study is the inclusion of labor ward nursing staff who are generally not the primary focus of KMC implementation. Therefore, the inclusion of labor ward nursing staff in future intervention would provide additional opportunities to introduce mothers to KMC and support mothers in using KMC.

## Conclusions

This current study revealed that nurses and midwives at Koja District Hospital lacked KMC training. Notably, 73.8% of all nurses/midwives have never received relevant training and exhibited inadequate general knowledge on KMC. Moreover, no SOPs existed, and there was the lack of a designated facility for KMC implementation throughout the time of this baseline study. Nonetheless, it should be emphasized that the provision of appropriate KMC facilities does not guarantee successful KMC implementation. Nurses, midwives, and all relevant healthcare providers must also possess the necessary knowledge and attitude to encourage and educate parents to practice and implement KMC.

## Recommendation

To improve KMC implementation in hospitals, effective interventions are needed. These should include training that incorporates KMC content into the nurse/midwife curriculum and extracurricular training. Also the provision of facilities that support KMC (such as couches or beds or a designated ward) coupled with on-site training and clear guidelines/SOPs for each relevant ward should also be taken into consideration. Engagement of key stakeholders (e.g., engaging the hospital director and management to obtain their support and commitment) is also a key factor in the success of KMC implementation, given their role in providing the necessary resources and ensuring optimum processes.

## Acknowledgments

We acknowledge all the research teams participating in this KMC project under the PEER project; the enumerators who helped execute the study and collect data; Trisari Anggondowati, Ph.D, for her input regarding this manuscript; all the respondents; and the authorities at Koja District Hospital who made this study possible. The authors also would like to thank Enago (www.enago.com) for the English language review.

## Author Contributions

**Conceptualization:** Asri Adisasmita, Yulia Izati, Septyana Choirunisa, Hadi Pratomo, Luzy Adriyanti.

**Formal analysis:** Asri Adisasmita, Septyana Choirunisa.

**Investigation:** Yulia Izati, Septyana Choirunisa.

**Methodology:** Asri Adisasmita, Yulia Izati, Septyana Choirunisa.

**Supervision:** Asri Adisasmita, Yulia Izati, Septyana Choirunisa.

**Writing – original draft:** Asri Adisasmita, Yulia Izati, Septyana Choirunisa, Hadi Pratomo, Luzy Adriyanti.

**Writing – review & editing:** Asri Adisasmita, Yulia Izati, Septyana Choirunisa, Hadi Pratomo, Luzy Adriyanti.

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
