## [Decision Letter · Decision Letter 0]

27 Oct 2020

PONE-D-20-28218

Knowledge, Attitudes, and Practices of Nursing Staff toward Kangaroo Mother Care at Koja Hospital, North Jakarta Municipality, Jakarta, Indonesia: A Cross-sectional Descriptive Study

PLOS ONE

Dear Dr. Adisasmita,

Thank you for submitting your manuscript to PLOS ONE. After careful consideration, we feel that it has merit but does not fully meet PLOS ONE’s publication criteria as it currently stands. Therefore, we invite you to submit a revised version of the manuscript that addresses the points raised during the review process.

We look forward to receiving your revised manuscript.

Kind regards,

Elisabete Alves

Academic Editor

PLOS ONE

Journal Requirements:

2. Please include additional information regarding the survey or questionnaire used in the study and ensure that you have provided sufficient details that others could replicate the analyses. For instance, please include additional details regarding the questionnaire content, development and validation.  If the questionnaire is not under a copyright more restrictive than CC-BY, please include a copy, in both the original language and English, as Supporting Information.

Reviewers' comments:

Reviewer's Responses to Questions

**Comments to the Author**

1. Is the manuscript technically sound, and do the data support the conclusions?

Reviewer #1: Yes

Reviewer #2: Yes

2. Has the statistical analysis been performed appropriately and rigorously? 

Reviewer #1: Yes

Reviewer #2: No

3. Have the authors made all data underlying the findings in their manuscript fully available?

Reviewer #1: Yes

Reviewer #2: No

4. Is the manuscript presented in an intelligible fashion and written in standard English?

Reviewer #1: Yes

Reviewer #2: Yes

5. Review Comments to the Author

Reviewer #1: The authors provide a detailed exploration of knowledge attitude and practices of nursing staff in relation to KMC at a single institution.

The study design is appropriate for the objective of the study

The background/introduction is succinct and provides a good synopsis of the topic including a local context.

The methods are well described. However, on line 70, the authors should consider sharing that this is the quantitative portion of a larger study that utilized both qualitative and quantitative methodology. As currently written, it appears unclear. A scale to describe the definition of low/moderate/high knowledge could have been presented in this section. This terminology appeared in the results section and is unclear. Simple descriptive statistics was appropriate for this inquiry.

Results: For all tables, if data is missing, an annotation should be made to signify such. Table 3 has 2 different data points that do not add up to the specified "n". The "n" for "attended training that included KMC" under perinatal ward does not add up to 29 like others and for the same question Postnatal ward does not add up to 21. The authors should maintain consistency in reporting data, presenting as raw number with percentage would be a preferred style. To minimize clutter, the authors should consider presenting Table 3 as a stacked column or bar chart.

Figure 1 Legend should be reviewed for grammar to ensure clarity. Line 197 "dan" vs. "and".

Line 204: they had a negative attitude or they negatively answered a question? Attitude is being described and not a correct or incorrect question? The authors should re-evaluate framing of this section.

Conclusion and Recommendations: accurately describe the findings

I would recommend a review of the manuscript by a language editing service to ensure better structure and grammar.

Reviewer #2: The paper “Knowledge, Attitudes, and Practices of Nursing Staff toward Kangaroo Mother Care at Koja Hospital, North Jakarta Municipality, Jakarta, Indonesia: A Cross-sectional Descriptive Study” is an interest study about the experience of nursing staff on KMC in an Indonesian hospital. The paper needs to be clearer about the innovativeness of its findings because there is a large number of relevant papers in the field tackling these issues.

Title:

- Authors should consider choosing a shorter title.

Introduction:

- Authors should consider to update the worldwide and Indonesian preterm birth rates available for 2014 at “Chawanpaiboon S, Vogel JP, Moller AB, Lumbiganon P, Petzold M, Hogan D, Landoulsi S, Jampathong N, Kongwattanakul K, Laopaiboon M, Lewis C, Rattanakanokchai S, Teng DN, Thinkhamrop J, Watananirun K, Zhang J, Zhou W, Gülmezoglu AM. Global, regional, and national estimates of levels of preterm birth in 2014: a systematic review and modelling analysis. Lancet Glob Health. 2019 Jan;7(1):e37-e46. doi: 10.1016/S2214-109X(18)30451-0. Epub 2018 Oct 30. PMID: 30389451; PMCID: PMC6293055.”

- Once the main objective of this study is to inform the design of larger interventions for improving KAP of KMC among nurses, why the authors did not consider to assess the needs of these professionals for implementing this model of care?

Methods:

- Authors should consider to provide the number and % of PT babies during data collection period.

- Authors should consider explaining the participants’ recruitment process. Is the participation rate 100%? Why?

- This sentence “Study findings are presented in text, tables, and figures” is redundant.

- The chi-squared test is not the most adequate test to use in most of the comparisons because of its assumption that the expected value in each cell is greater than 5.

Results:

- Page 6, lines 122-123: This information may be at methods section.

Kangaroo mother care training

- Values from table 2 are very small which compromise the value of the statistical test.

Table 3

- Is it correct to present a p value for a statistical test with 0 individuals in one of the cells?

- The percentages of the table are not in column neither in line! It is very confusing for the readers. Could the authors explain their option?

Discussion:

- The discussion section is confusing and presents a circular way of thinking. Author should consider structuring it in a better way.

- Page 12, lines 231-234: Authors should discuss the implications of nurses implementing KMC without training. They presume that this is a good thing but there are a lot of negative implications behind that, considering the results on KMC knowledge.

- Page 14, lines 276-284: A reference is needed.

- Pages 14-15, lines 285-295: Authors discussed some findings not reported in the results, namely “the lack of facilities or supporting policy and equipment”. In fact, nurses need this type of facilities to better implement KMC, not only adequate education as the authors suggest at the end of this paragraph.

- Authors should consider do not limit the discussion around the need for training on KMC. There are a bunch of hospital policies that need to be implemented in order to guarantee the basic conditions for a good KMC implementation.

References:

- Some references need to be updated.

6. PLOS authors have the option to publish the peer review history of their article (what does this mean?). If published, this will include your full peer review and any attached files.

Reviewer #1: No

Reviewer #2: No

---

## [Author Response · Author response to Decision Letter 0]

26 Feb 2021

Dear Reviewers, 

Thank you for the useful inputs and review. We have revised our manuscript based on input from reviewers and include responses (blue color text) to reviewers' comments as follows:

Reviewer #1: 

1. The authors provide a detailed exploration of knowledge attitude and practices of nursing staff in relation to KMC at a single institution.

The study design is appropriate for the objective of the study

The background/introduction is succinct and provides a good synopsis of the topic including a local context.

The methods are well described. However, on line 70, the authors should consider sharing that this is the quantitative portion of a larger study that utilized both qualitative and quantitative methodology. 

We have added in the methods section line 86-88 (of manuscript file without track changes) mentioning that this quantitative study is a portion of a larger study both utilize qualitative and quantitative methodology: “The study is part of a larger research project that used quantitative and qualitative methods, but this part only focused on the quantitative part of the research”.

2. As currently written, it appears unclear. A scale to describe the definition of low/moderate/high knowledge could have been presented in this section. This terminology appeared in the results section and is unclear. 

We added description of low/moderate/high knowledge in line 135-138 (of manuscript file without track changes) in methods section: “General knowledge, knowledge on benefit, and attitude score were grouped into four score categories (scores 0 to 25 = very low, 25 to below 50 = low, 50 to below 75 = moderate, and 75 to 100 = good/high).

Simple descriptive statistics was appropriate for this inquiry.

3. Results: For all tables, if data is missing, an annotation should be made to signify such. 

We added annotation under table 1: “aPercentages that do not add up to 100% was due to missing data (i.e., one respondent from the postnatal ward did not provide an answer for years of experience working in the current unit)” in line 155-157 of file Manuscript (without track changes).

4. Table 3 has 2 different data points that do not add up to the specified "n". The "n" for "attended training that included KMC" under perinatal ward does not add up to 29 like others and for the same question Postnatal ward does not add up to 21.

We edited table 3 into the current format. The previous table 3 had information on good knowledge (assuming that the rest are not good knowledge). In the current table 3, we provide information including all levels i.e. low, moderate, and good. So the numbers will add up to 29 or 21. 

5. The authors should maintain consistency in reporting data, presenting as raw number with percentage would be a preferred style. To minimize clutter, the authors should consider presenting Table 3 as a stacked column or bar chart.

We already edited the reporting data consistently using raw numbers and percentages. We did consider using a graph, but we decide to use table format for table 3 to incorporate much information we want to describe.

6. Figure 1 Legend should be reviewed for grammar to ensure clarity. Line 197 "dan" vs. "and".

We have already edited it in line 214.

7. Line 204: they had a negative attitude or they negatively answered a question? 

We mean negatively answered the question. We then edited the sentence into: “When asked about KMC implementation among infants weighing 1,000–1,800 g, 40% of respondents from the labor ward were not in favor of KMC implementation”. in line 219-221 of the file Manuscript (without track changes)

8. Attitude is being described and not a correct or incorrect question? The authors should re-evaluate framing of this section.

We have changed the sentence into: “No differences were observed after stratifying respondents’ scores according to age group and training experience, and 100% of nurses had a good attitude (score > 75) about KMC (Table 3)” in line 221-223 of file Manuscript (without track changes)

9. Conclusion and Recommendations: accurately describe the findings

I would recommend a review of the manuscript by a language editing service to ensure better structure and grammar.

We have used language editing service to proofread and edit our manuscript. 

Reviewer #2:

10. The paper “Knowledge, Attitudes, and Practices of Nursing Staff toward Kangaroo Mother Care at Koja Hospital, North Jakarta Municipality, Jakarta, Indonesia: A Cross-sectional Descriptive Study” is an interest study about the experience of nursing staff on KMC in an Indonesian hospital. The paper needs to be clearer about the innovativeness of its findings because there is a large number of relevant papers in the field tackling these issues.

Title:

- Authors should consider choosing a shorter title.

We have edited our title into: “Kangaroo mother care knowledge, attitude, and practice among nursing staff in a hospital in Jakarta, Indonesia”.

11. Introduction:

- Authors should consider to update the worldwide and Indonesian preterm birth rates available for 2014 at “Chawanpaiboon S, Vogel JP, Moller AB, Lumbiganon P, Petzold M, Hogan D, Landoulsi S, Jampathong N, Kongwattanakul K, Laopaiboon M, Lewis C, Rattanakanokchai S, Teng DN, Thinkhamrop J, Watananirun K, Zhang J, Zhou W, Gülmezoglu AM. Global, regional, and national estimates of levels of preterm birth in 2014: a systematic review and modelling analysis. Lancet Glob Health. 2019 Jan;7(1):e37-e46. doi: 10.1016/S2214-109X(18)30451-0. Epub 2018 Oct 30. PMID: 30389451; PMCID: PMC6293055.”

We have updated our references in line 46-48 of file Manuscript (without track changes)

12. - Once the main objective of this study is to inform the design of larger interventions for improving KAP of KMC among nurses, why the authors did not consider to assess the needs of these professionals for implementing this model of care?

We did consider assessing the need of this professional for implementing this model of care through our qualitative study (part of the larger study). 

From our formative research which is not published yet, the findings were included in our project report (reference no.9), we assessed the needs of the nursing staff, pediatrician, and management, in having intervention (including knowledge, skill, among others) to improve KMC implementation in the hospital. 

In addition, we also cited the qualitative findings regarding this issue in the discussion section although not straightforward in line 296-298 “However, to educate and support the mother/family in KMC, the nursing staff needs formal training to improve their skill and confidence in supporting KMC [19]”

13. - Authors should consider explaining the participants’ recruitment process. Is the participation rate 100%? Why?

Yes, the participation rate was 100%. We provide information that all nursing staff willing to participate in line 119 of the file Manuscript (without track changes), although not all of them provide complete data (there is missing data in years of working in current unit). 

- This sentence “Study findings are presented in text, tables, and figures” is redundant.

We have deleted the sentence above to avoid redundancy.

14. - The chi-squared test is not the most adequate test to use in most of the comparisons because of its assumption that the expected value in each cell is greater than 5.

We added Fisher Exact test if the expected value in a cell is less than 5 in line 141

15. Results:

- Page 6, lines 122-123: This information may be at methods section.

We deleted this sentence in Results section and move it to Methods section in line 118-119.

16. Kangaroo mother care training

- Values from table 2 are very small which compromise the value of the statistical test.

Table 3

- Is it correct to present a p value for a statistical test with 0 individuals in one of the cells?

The statistical output is presented below:

KMC_training_baseline * Ruang Crosstabulation

 Ruang Total

 Perinatology Labor Rooming-in 

KMC_training_baseline both specific and KMC included training Count 2 2 1 5

 % within Ruang 6.9% 13.3% 4.8% 7.7%

 specific KMC training Count 2 0 1 3

 % within Ruang 6.9% 0.0% 4.8% 4.6%

 KMC included training Count 5 3 1 9

 % within Ruang 17.2% 20.0% 4.8% 13.8%

 not participate in any Count 20 10 18 48

 % within Ruang 69.0% 66.7% 85.7% 73.8%

Total Count 29 15 21 65

 % within Ruang 100.0% 100.0% 100.0% 100.0%

Chi-Square Tests

 Value df Asymptotic Significance (2-sided)

Pearson Chi-Square 4.401a 6 .623

Likelihood Ratio 5.323 6 .503

Linear-by-Linear Association .705 1 .401

N of Valid Cases 65 

a. 9 cells (75.0%) have expected count less than 5. The minimum expected count is .69.

17. - The percentages of the table are not in column neither in line! It is very confusing for the readers. Could the authors explain their option?

We have edited table 3 

18. Discussion:

- The discussion section is confusing and presents a circular way of thinking. Author should consider structuring it in a better way.

We have attempted to revise the discussion section.

19. - Page 12, lines 231-234: Authors should discuss the implications of nurses implementing KMC without training. They presume that this is a good thing but there are a lot of negative implications behind that, considering the results on KMC knowledge.

We added the sentence in the discussion: “Also, the lack of training could result in conflicting knowledge on the timing and duration of KMC [24,35], which could lead to adverse consequences such as mortality [36] especially among less stable infants” in line 256-258.

21. Page 14, lines 276-284: A reference is needed.

We have edited the sentences and provide references that could be seen in line 298-302.

22. - Pages 14-15, lines 285-295: Authors discussed some findings not reported in the results, namely “the lack of facilities or supporting policy and equipment”. In fact, nurses need this type of facilities to better implement KMC, not only adequate education as the authors suggest at the end of this paragraph.

We added the reference regarding lack of facilities or supporting policy and equipment in line 306-308. The information was gathered during formative research (larger study).

23. - Authors should consider do not limit the discussion around the need for training on KMC. There are a bunch of hospital policies that need to be implemented in order to guarantee the basic conditions for a good KMC implementation.

We have added in line 317-319: “However, in addition to adequate education, adequate equipment/facilities as well as SOPs and hospital policy supporting KMC implementation are essential for successful KMC in the hospital”

24. References:

- Some references need to be updated.

We have updated the references

---

## [Decision Letter · Decision Letter 1]

8 Apr 2021

PONE-D-20-28218R1

Kangaroo mother care knowledge, attitude, and practice among nursing staff in a hospital in Jakarta, Indonesia

PLOS ONE

Dear Dr. Adisasmita,

Thank you for submitting your manuscript to PLOS ONE. After careful consideration, we feel that it has merit but does not fully meet PLOS ONE’s publication criteria as it currently stands. Therefore, we invite you to submit a revised version of the manuscript that addresses the points raised during the review process.

We look forward to receiving your revised manuscript.

Kind regards,

Elisabete Alves

Academic Editor

PLOS ONE

Journal Requirements:

Reviewers' comments:

Reviewer's Responses to Questions

**Comments to the Author**

1. If the authors have adequately addressed your comments raised in a previous round of review and you feel that this manuscript is now acceptable for publication, you may indicate that here to bypass the “Comments to the Author” section, enter your conflict of interest statement in the “Confidential to Editor” section, and submit your "Accept" recommendation.

Reviewer #2: (No Response)

2. Is the manuscript technically sound, and do the data support the conclusions?

Reviewer #2: Yes

3. Has the statistical analysis been performed appropriately and rigorously? 

Reviewer #2: Yes

4. Have the authors made all data underlying the findings in their manuscript fully available?

Reviewer #2: Yes

5. Is the manuscript presented in an intelligible fashion and written in standard English?

Reviewer #2: Yes

6. Review Comments to the Author

Reviewer #2: I would like to thank the authors for their work in answering reviewers' questions.

I think the paper still has some minor issues that should be revised, namely inconsistencies in showing the results in the tables. In table 2 is still missing the row for n(%) as shown in table 1. In table 3 authors reported the p values inconsistently - with and without the 0 before the comma. I think authors should revise all tables again in order to present the results in a consistent way.

7. PLOS authors have the option to publish the peer review history of their article (what does this mean?). If published, this will include your full peer review and any attached files.

Reviewer #2: No

---

## [Author Response · Author response to Decision Letter 1]

18 May 2021

Our response:

We would like to thank the reviewer for giving the input for the manuscript. In the revised manuscript:

• We added n (%) in table 2 

• We edited p-values in table 3 by adding 0 before the comma

• We revised the tables’ layout to make it consistent and easier to understand

---

## [Editor Report · Decision Letter 2]

21 May 2021

Kangaroo mother care knowledge, attitude, and practice among nursing staff in a hospital in Jakarta, Indonesia

PONE-D-20-28218R2

Dear Dr. Adisasmita,

We’re pleased to inform you that your manuscript has been judged scientifically suitable for publication and will be formally accepted for publication once it meets all outstanding technical requirements.

Kind regards,

Elisabete Alves

Academic Editor

PLOS ONE
---

## [Editor Report · Acceptance letter]

27 May 2021

PONE-D-20-28218R2 

Kangaroo mother care knowledge, attitude, and practice among nursing staff in a hospital in Jakarta, Indonesia 

Dear Dr. Adisasmita:

I'm pleased to inform you that your manuscript has been deemed suitable for publication in PLOS ONE. Congratulations! Your manuscript is now with our production department. 

Kind regards, 

on behalf of

Dr. Elisabete Alves 

Academic Editor

PLOS ONE